# Berberine Reveals Anticoccidial Activity by Influencing Immune Responses in *Eimeria acervulina*-Infected Chickens

**DOI:** 10.3390/biom15070985

**Published:** 2025-07-10

**Authors:** Binh T. Nguyen, Bujinlkham Altanzul, Rochelle A. Flores, Honghee Chang, Woo H. Kim, Suk Kim, Wongi Min

**Affiliations:** 1College of Veterinary Medicine & Institute of Animal Medicine, Gyeongsang National University, Jinju 52828, Republic of Korea; thanhbinhcnty@gmail.com (B.T.N.); bvjbvjka@gmail.com (B.A.); floresrochellea@gmail.com (R.A.F.); woohyun.kim@gnu.ac.kr (W.H.K.); kimsuk@gnu.ac.kr (S.K.); 2Faculty of Animal Science and Veterinary Medicine, Thai Nguyen University of Agriculture and Forestry, Thai Nguyen 24119, Vietnam; 3 Division of Animal Science, Gyeongsang National University, Jinju 52828, Republic of Korea; hhchang@gnu.ac.kr; 4Hoxbio, Business Center, Gyeongsang National University, Jinju 52828, Republic of Korea

**Keywords:** chickens, coccidiosis control, *E. acervulina*, berberine, evaluation of parameters, transcriptomic analysis

## Abstract

Serious enteric disease caused by seven species of *Eimeira* continues to cause significant economic damage to the poultry industry. *E. acervulina* is one of the most widespread strains in farms and has a significant impact on chicken weight loss. Currently, the use of anticoccidial agents to suppress the occurrence of coccidiosis in farms is considerably restricted due to public health and environmental pollution issues. It is important to understand the protective immunity of the host against *Eimeria* infections with regard to natural products that could be used as alternatives to anticoccidial agents. Berberine chloride is known for its various biological functions, including its anti-parasite activity. However, its impact on intestinal morphology and immune-related activity in broilers infected with *Eimeria* still remains unclear. The aim of this study is to evaluate the anticoccidial effects of a berberine-based diet in broilers infected with *E. acervulina* and to monitor the host immune phenomenon using transcriptomic analysis. Administration of berberine to chickens infected with *E. acervulina* significantly reduced fecal oocyst production and intestinal lesion scores, and increased duodenal villus height, indicating anticoccidial activity and positive effects on intestinal morphology. Transcriptomic analysis of chickens infected with *E. acervulina* generally observed the down-regulation of metabolism-related genes and the up-regulation of cell integrity-related genes at day 4 post-infection. At day 6 post-infection, an increase in immune-related genes and cellular-homeostasis-related genes was generally observed. Berberine-treated and *E. acervulina*-infected chickens showed cytokine-cytokine receptor interaction in the second term in a Kyoto Encyclopedia of Genes and Genomes pathway analysis at day 4 post-infection, but not in chickens infected with *E. acervulina* alone, suggesting host immune changes induced by berberine. These results suggest that berberine, which exhibits anticoccidial effects, may have therapeutic and/or prophylactic potential in protecting the host from infectious and economic-loss-causing diseases, such as *Eimeria* infection.

## 1. Introduction

The poultry industry is one of the most valuable sources of animal protein globally, providing meat as well as eggs [1]. There are a variety of infectious pathogens that have a wide range of negative impacts on poultry health [2]. Among these, chicken coccidiosis is an important gut disease, caused by protozoan parasites of the genus *Eimeria* belonging to the Apicomplexa phylum. There are seven distinct *Eimeria* species (*E. acervulina*, *E. brunetti*, *E. maxima*, *E. mitis*, *E. necatrix*, *E. praecox*, and *E. tenella*) that invade the gastrointestinal epithelial tissues in a region-specific manner and cause a wide range of clinical manifestations in infected hosts, from asymptomatic to death [3,4]. Therefore, this disease is considered to be one of the most economically important diseases in the poultry industry because of its negative impact on growth performance, feed conversion ratio (FCR), susceptibility to secondary pathogens, and mortality [5,6]. The annual global loss due to chicken coccidiosis was recently estimated at more than USD 14.5 billion [7].

Management of coccidiosis has focused on several strategies, including farm-level management, vaccination, and use of anticoccidials [4,8]. Anticoccidial drugs have been used for more than 60 years, typically as additives to feed or drinking water, which has led to the emergence of antibiotics-resistant strains of *Eimeria* species in the field that threaten the continued growth of the poultry industry [9]. Additionally, drug residues in poultry products, such as meat and eggs, have been of concern as potential human health risks, leading to calls for additional global regulations to limit or ban their use [1,10]. Although live or attenuated vaccines are effective in reducing this disease in commercial poultry industries, concerns exist regarding safety and increased production costs [8,9]. Commercial vaccines are relatively expensive because the process of producing various strains is difficult and tedious. There is little or no cross-protection to challenge infection with a heterologous *Eimeria* species, and even infection with a different strain of the same species is poorly protected in some cases [11,12]. The feces in poultry farms contains several *Eimeria* species [11,13]. Thus, live commercial vaccines contain multiple *Eimeria* species with controlled numbers of oocysts for each *Eimeria* species, and all vaccine strains must be propagated in chickens without cross-contamination under strict specific pathogen-free conditions [9,12]. A main consideration when administering live vaccines is that fecal–oral recycling of the parasites is required to induce sufficient active immunity to protect chickens from pathogenic challenge by most *Eimeria* species, and, consequently, there is a high risk of causing clinical coccidiosis [12,14]. Additionally, live vaccines may increase the outbreak of bacterial enteritis in the absence of growth promoters [10]. Therefore, the application of anticoccidials will continue until vaccines become more sophisticated and relatively cost-competitive.

Natural plant-based products are promising resources for identifying potential therapeutic and prophylactic agents to parasites, and one of these is berberine [15,16,17,18]. Berberine is a bitter-tasting isoquinoline alkaloid that fluoresces yellow under ultraviolet light. It is obtained from the stems and roots of numerous types of medicinal plants, including *Berberis*, *Hydrastis canadensis*, and *Coptidis rhizoma*, which have been used in traditional Chinese medicine to treat gut disorders [19,20]. Berberine also has various biological activities, including antiarrhythmic, antiplatelet, antidiabetic, antiviral, antibacterial, and anti-inflammatory pharmacological effects [19,20,21,22].

Berberine has recently received increasing attention for its anticoccidial effects. In mice infected with *E. papillate*, berberine induced anticoccidial effects by impairing apoptotic processes, reducing inflammatory and oxidative stress responses, and activating local innate immune activities [16]. Additionally, berberine reduced the viability and sporulation rate of *E. papillate* unsporulated oocysts in an in vitro assay and the number of zygotes and developing oocysts within gut villi in *E. papillate*-infected mice [23]. Al-Quraishy et al. [24] reported that berberine exhibited a great enhancement in nutrient homeostatic status and also restored jejunal carbohydrate content in *E. papillate*-infected mice. In poultry, the anticoccidial activity of berberine was mainly monitored in chickens infected with *E. tenella* [17,25,26]. Compared to *E. tenella*-infected and berberine-untreated chickens, berberine-treated chickens had reduced oocyst production, increased bodyweight gain, and lower feed conversion ratio [25,26]. A propylene glycol extract of *Allium sativum* and *Thymus serpyllum* was effective in decreasing intestinal lesions caused by *E. acervulina*, but not *E. tenella*, suggesting that there may be differences in susceptibility to natural products depending on the *Eimeria* species [27]. Our previous study tested the anticoccidial effect of berberine using five different *Eimeria* species, and reduction in fecal oocyst production was identified in all of the *Eimeria* species tested [15]. However, the anticoccidial mechanism of action of berberine in chickens infected with *Eimeria* remains to be elucidated. Accordingly, this study was performed to further investigate the anticoccidial mechanisms of berberine-based diets in *E. acervulina*-infected chickens using comparative transcriptome profiling along with the assessment of three parameters: fecal oocyst shedding, intestinal lesion score, and histopathological findings.

## 2. Materials and Methods

### 2.1. Ethics Statement

All animal experiments were approved by the Institutional Animal Care and Use Committee (GNU-201116-C0084) and were conducted in accordance with the Gyeongsang National University Guidelines for the Care and Use of Experimental Animals. Specific endpoint criteria were established to ensure animal welfare, including immediate euthanasia by atlanto-occipital dislocation for birds exhibiting tremors, severe bodyweight loss, or unresponsiveness to stimuli. The remaining birds were euthanized at predetermined time points. Every effort was made to minimize the bird’s pain and suffering.

### 2.2. Animals and Infection

Ross 308 broiler chickens, obtained from a local hatchery (Samhwa, Hongseong, Republic of Korea), were housed in wire cages for 2 weeks in a temperature-controlled environment. Throughout the study, the chickens were provided with continuous lighting, unrestricted access to antibiotic-/anticoccidial-free feed, and clean tap water. The composition of the broiler feed is provided (Appendix A). At 2 weeks old, the chickens were randomly divided into three groups (*n* = 50): group 1 (NC) consisted of uninfected and untreated chickens, group 2 (EA) included chickens infected with *E. acervulina* but untreated, and group 3 (EA+BBR) consisted of chickens infected with *E. acervulina* and simultaneously fed a 2 g berberine (BBR)/kg-supplemented diet. Chickens in groups 2 and 3 were infected with 1 mL of 100,000 sporulated oocysts of *E. acervulina* per bird by oral gavage. The duodenal tissues were obtained from three chickens per group at 4 days post-infection (dpi) and 6 dpi for a subsequent transcript expression analysis using RNA-seq. Specifically, 1~2 cm of the duodenum was excised, placed in a 5 mL Eppendorf tube, snap-frozen in liquid nitrogen, and stored at −70 °C until processing. Feces were collected from 5 to 9 dpi and homogenized in a blade grinder to assess fecal oocyst shedding. Samples were diluted in saturated NaCl, and numbers of the oocysts were counted using an optical microscope in a McMaster counting chamber. The oocyst numbers were calculated from the average of three counts per sample. The total oocyst numbers were calculated as follows: oocyst count × dilution factor × (fecal sample volume ÷ counting chamber volume). After careful removal of the intestinal contents, three blinded observers independently assigned a numerical lesion score from 0 (no gross lesions) to 4 (representing extensive and severe hemorrhage or lesions) to each segment based on a previously described scoring technique [28]. Bodyweight gain was recorded at day 9 after post-infection. Figure 1 is a schematic diagram of the experimental design. Berberine hydrochloride was purchased from Daejung Chemicals & Metals (Siheung, Korea) at the highest available purity (≥98%).

### 2.3. Histopathological Examination

At 6 dpi, duodenal tissues (*n* = 5) were obtained from each group for histopathological examination. The intestinal tissues from each group were fixed in 10% neutral-buffered formalin solution, dehydrated using a graded series of alcohol and xylene, and embedded in paraffin to create tissue blocks. Tissue sections of 4 µm from each sample were stained with hematoxylin and eosin (H&E) and examined for histological evaluation using an optical microscope at 40× magnification. The crypt depth (CD) was determined by measuring the distance from the bottom of the mucosa (bottom of the crypt) to the top of the crypt. Similarly, the villus height (VH) was measured from the bottom of the mucosa to the top of the villi.

### 2.4. Library Construction and Transcriptome Sequencing

Total RNA extraction from 18 intestinal duodenum tissues was performed using TRIzol reagent (Invitrogen, Carlsbad, CA, USA), following the manufacturer’s recommendations. RNA-seq library construction was performed using TruSeq DNA PCR (eGnome, Seoul, Republic of Korea), and sequencing was performed on an Illumina NovaSeq 6000 platform (eGnome, Seoul, Republic of Korea). FastQC (v0.11.7) was used to filter raw reads in FastQ files to obtain high-quality reads. Trimmomatic (v0.38) was used to trim adapter sequences with quality scores less than 36 from the raw reads. The final high-quality reads were mapped to the *Gallus gallus* (GRCg6a; GCF_00002315.6) reference genome using HISAT2 (v2.1.0), and StringTie (v2.1.3b) was used to quantify reads in terms of the genomic features.

### 2.5. Transcriptome Data Analysis

Analysis of differential gene expression was performed using DESeq2 (v1.38.3). Significantly differentially expressed genes (DEGs) with Log_2_ fold change (log_2_FC) ≥ 2 and a false discovery rate (FDR)-adjusted *p*-value < 0.05 were defined to be regulated differentially in the three comparison groups (EA vs. NC (control); EA+BBR vs. NC; EA+BBR vs. EA). To further analyze the biological processes associated with DEGs, gene ontology (GO)-based enrichment analysis was tested using a one-sided hypergeometric test in g:Profiler (ve111_eg58_p18_f463989d, https://biit.cs.ut.ee/gprofiler/, accessed on 6 November 2024), which provided information on the relevant biological process (BP), molecular function (MF), and cellular component (CC) domains. Kyoto Encyclopedia of Genes and Genomes (KEGG) functional-enrichment analysis was subsequently performed using modified Fisher’s exact test (https://www.genome.jp/kegg/pathway.html, accessed on 6 November 2024). A search tool for retrieving interacting genes/proteins (STRING v12.0, https://string-db.org/, accessed on 7 April 2025) was used for the protein–protein interaction (PPI) network analysis of DEGs. The distribution, distance, and overall relationships of the samples were visualized using the DEG viewer system (v2.0, https://degviewer.macrogen.com/, accessed on 25 March 2025) for multidimensional scaling plots, volcano plots, and heatmaps.

### 2.6. Statistical Analysis

The statistical software package Instat (GraphPad Prism (v5.03), Boston, MA, USA) or Student’s *t*-test was used for data analysis. Data analyses were performed with one-way ANOVA followed by a nonparametric statistical test using the Kruskal–Wallis test, and means were compared using Dunn’s multiple comparison test. Data were expressed as mean ± standard error values. Differences were considered statistically significant at *p* < 0.05.

## 3. Results

### 3.1. Berberine Positively Influences Gut Morphology in Chickens Infected with E. acervulina

Our previous study showed that berberine treatment significantly decreased oocyst production in chickens infected with five different *Eimeria* species compared to untreated chickens, but there were differences in sensitivity to berberine. *E. acervulina* and *E. praecox* were generally the most sensitive species, whereas *E. maxima* was the least sensitive species [15]. Of the most sensitive species, *E. acervulina* is the most prevalent species in the poultry industry [11,13]. Thus, this study was carried out with *E. acervulina*-infected chickens.

At 6 dpi, the lesion score was significantly reduced in the berberine-treated and infected (EA+BBR) group (3.8 ± 0.18) compared to the untreated and infected (EA) group (2.0 ± 0.28) (Figure 2A). There was a significant difference in fecal oocyst production between the EA (6.1 × 10^8^ ± 5.5 × 10^6^) and EA+BBR groups (5.2 × 10^8^ ± 7.1 × 10^6^) (Figure 2B). The untreated and uninfected control (NC) group showed no lesion score and fecal oocyst shedding. Bodyweight gain was significantly decreased in the EA+BBR group compared to the NC and EA groups, but no difference was observed between the NC and EA groups (Appendix A). In the histopathological analysis (Figure 2C–H), berberine treatment showed an increase in villus height (VH) without an effect on crypt depth (CD) (Figure 2C–E). A significant reduction in VH was observed in the EA group, but this was compensated for by berberine treatment. The VH was reduced by approximately 51.5% (829 ± 25.9 μm, *p* < 0.001) in the EA group and increased by 1.0% (1791 ± 74.8 μm) in the EA+BBR group, respectively, compared to the NC group (1709 ± 44.1 μm) (Figure 2F). The CD increased by approximately 75.8% (459 ± 17.2 μm, *p* < 0.001) in the EA group and 70.1% (446 ± 29.0 μm, *p* < 0.001) in the EA+BBR group, respectively, compared to the NC group (261 ± 16.3 μm) (Figure 2G). Compared to the VH to CD ratio in the NC group (7.4 ± 0.6), there was a significant decrease from 75.5% to 41.9% in the EA group (1.8 ± 0.05, *p* < 0.001) and the EA+BBR group (4.3 ± 0.4, *p* < 0.05), indicating that berberine treatment increased the VH and the VH to CD ratio compared to the EA group infected with *E. acervulina* alone.

### 3.2. Comparative Transcriptomic Analysis

To elucidate the roles of berberine and its gene expression profile, duodenal tissues obtained at 4 and 6 dpi were utilized in a gene expression analysis using RNA-seq. The detailed sequencing information for each sample are presented (Appendix A). The sequencing generated an average number of 48.8 M total reads (range: 38.8 to 56.7 M) and of 43.6 M mapped read (range: 34.2 to 51.6 M). The average GC content and Q30 of these clean reads were 49.9 ± 0.1% and 91.3 ± 0.1%, respectively (Appendix A).

Principal component analysis (PCA) was used to determine the similarity of expression levels among the transcriptomes (Figure 3A–C). The resulting plot at 4 dpi revealed no significant differences between the three groups (PERMANOVA, *p* = 0.087) (Figure 3A). At 6 dpi, principal component 1 and 2 explained 29.6% and 25.0% of the variable variance, respectively, and the cumulative contribution rate was 54.6%. The resulting plot revealed significant differences between the three groups (PERMANOVA, *p* = 0.003) (Figure 3B). Additionally, the plots including both the 4-day and 6-day groups showed a significant difference (PERMANOVA, *p* = 0.001) (Figure 3C). Principal components 1 and 2 explained 30.4% and 16.8% of the variance in the variables, respectively. The hierarchical clustering was estimated based on Euclidean distance and is visualized using a heatmap and dendrogram. Similarly to the PCA plots, the heatmap consisting of 1267 DEGs showed higher similarity between the EA+BBR and NC groups at 4 dpi. The EA group formed a separate clade (Figure 3D). At 6 dpi (Figure 3E), the heatmap constituting 961 DEGs showed higher similarity between EA and EA+BBR groups. The NC group formed a separate clade, suggesting that BBR treatment induced significant changes in the transcriptome during the course of *Eimeria* infection. A total of 1267 DEGs at 4 dpi were identified in three combinations, of which 643 were up-regulated and 624 were down-regulated. A total of 961 DEGs at 6 dpi were identified in three combinations, and, of these, 491 were up-regulated and 470 were down-regulated (Table 1).

Volcano plots depicting the DEGs in each dataset are illustrated in Figure 4. Compared to days 4 (441 DEGs) and 6 (572 DEGs) samples infected with *E. acervulina* only (Figure 4A,D), berberine treatment resulted in a 74.7% decrease (112 DEGs) in day 4 samples (Figure 4B) and a 51.6% decrease (277 DEGs) in day 6 samples (Figure 4E). In comparison of EA+BBR and EA groups, increased DEGs were observed at 4 dpi (Figure 4C), and decreased DEGs were observed at 6 dpi (Figure 4F).

### 3.3. GO and KEGG Pathway Analyses

#### 3.3.1. Pathway Analysis in *E. acervulina* Infection

According to GO function, all DEGs (Appendix A) were routinely classified into 3 categories: biological processes (BPs), molecular functions (MFs), and cellular components (CCs). The most significantly enriched GO terms between the EA and NC groups (EA vs. NC) at 4 dpi (Figure 5A–C) were the small-molecule metabolic process and transmembrane transport in BPs, transporter activity and oxidoreductase activity in MFs, and cell periphery and plasma membrane in CCs. The main KEGG pathways (Figure 6A) were cytoskeleton in muscle cells, the calcium signaling pathway, neuroactive ligand–receptor interaction, retinol metabolism, metabolism of xenobiotics by cytochrome P450, drug metabolism cytochrome P450, peroxisome, PPAR (peroxisome proliferator-activated receptor) signaling pathway, and the mitogen-activated protein kinase (MAPK) signaling pathway. At 6 dpi (Figure 5D–F), the most significantly enriched GO terms were the multicellular organismal process and cell surface receptor signaling pathway in MP, receptor ligand activity, signaling receptor activity and chemokine and cytokine activities in MFs, and extracellular region and cell periphery in CCs. The main KEGG pathways (Figure 6B) were neuroactive ligand–receptor interaction, ECM (extracellular matrix)–receptor interaction, cytokine–cytokine receptor interaction, the calcium signaling pathway, the MAPK signaling pathway, and the PPAR signaling pathway. The expression of interferon-γ (INF-γ), which is used as an important indicator in *Eimeria* infection [29,30], was up-regulated significantly at 6 dpi (Appendix A). In the overall gene expression analyses, the down-regulation of metabolism-related genes and up-regulation of cell integrity-related genes were generally observed at 4 dpi. At 6 dpi, an increase in immune-related genes and cell homeostasis-related genes was generally observed.

#### 3.3.2. Pathway Analysis in BBR Treatment and *E. acervulina* Infection

The most significantly enriched GO terms in the EA+BBR group compared to the NC group (EA+BBR group vs. NC group) at 4 dpi (Figure 5A–C) were L-serine biosynthetic process, monocyte chemotaxis and lymphocyte chemotaxis in BP, signaling receptor regulator activity and receptor ligand activity in BFs, and extracellular region and extracellular space in CCs. The main KEGG pathways (Figure 6A) were neuroactive ligand–receptor interaction, cytokine–cytokine receptor interaction, the MAPK signaling pathway, retinol metabolism, ECM–receptor interaction, and glycolysis/gluconeogenesis. At 6 dpi (Figure 5D–F), the most significantly enriched terms were multicellular organismal processes and immune system processes in BPs, signaling receptor binding and receptor ligand activity in BFs, and extracellular region and extracellular space in CCs. The main KEGG pathways (Figure 6B) were neuroactive ligand–receptor interaction, cytokine–cytokine receptor interaction, phagosome, the MAPK signaling pathway, and the PPAR signaling pathway.

In the EA+BBR group, compared to the EA group (EA+BBR group vs. EA group), at 4 dpi (Figure 5A–C), the most significantly enriched GO terms were small-molecule metabolic processes and organic hydroxy compound metabolic processes in BPs, transporter activity and transmembrane transporter activity in MFs, and membrane and cell periphery in CCs. The main KEGG pathways (Figure 6A) were the PPAR signal pathway, peroxisome, cytokine–cytokine receptor interaction, the calcium signaling pathway, and drug metabolism (cytochrome p450) (Figure 5B). At 6 dpi (Figure 5D–F), the most significantly enriched terms were multicellular organismal processes and cell–cell signaling in BPs, signaling receptor binding and iron ion binding in MFs, and extracellular region and cell periphery in CCs. The main KEGG pathways (Figure 6B) at 6 dpi were neuroactive ligand–receptor interaction, cytokine–cytokine receptor interaction, phagosome, the PPAR signal pathway, and the transforming growth factor (TGF)-β signaling pathway. Unlike *E. acervulina* infection only, a global gene expression analysis showed that berberine treatment generally stimulated the expression of immune-related genes at both 4 and 6 dpi (Figure 4 and Figure 5).

### 3.4. Effect of Berberine on Host Immune Changes in E. acervulina Infection

At 4 dpi, the KEGG pathway analysis results show that the cytokine–cytokine receptor interaction term was not classified into the top 10 terms in *E. acervulina* infection (EA vs. NC), although the cytokine–cytokine receptor interaction term included eleven DEGs, including five up-regulated genes and six down-regulated genes (Appendix A). As a result of berberine treatment in infected chickens (EA+BBR vs. NC), the cytokine–cytokine receptor interaction term was ranked second with eleven DEGs, including ten up-regulated genes and one down-regulated gene (Appendix A). Up-regulated genes included C-C motif chemokine ligand 21 (CCL21), growth differentiation factor 15 (GDF15), the tumor necrosis factor receptor superfamily, member 10b (TNFRSF10B), TNF receptor superfamily member 6b (TNFRSF6B), C-C motif chemokine ligand 4 (CCL4), interleukin 8-like 1 (IL8L1), interleukin 8-like 2 (IL8L2), tumor necrosis factor superfamily member 15 (TNFSF15), interleukin 22 (IL22), and C-C motif chemokine ligand 19 (CCL19). One down-regulated gene was C-C motif chemokine receptor 9 (CCR9) (Appendix A). Thus, the DEGs associated with cytokine–cytokine receptor interaction were further analyzed using a protein–protein interaction (PPI) network and heatmap (Figure 7). The PPI network of 25 DEGs at 6 dpi indicated a high degree of interaction between IFN-γ and chemokine-related genes (Figure 7A and Appendix A). As shown in Figure 7B, the heatmap shows that the EA+BBR group had an increase in DEGs related to the cytokine–cytokine receptor interaction term at 4 dpi compared to the EA group. This increase was mainly observed at 4 dpi rather than 6 dpi. The expression levels of three mRNAs, IL-22, IFN-γ, and C-C motif chemokine ligand 20 (CCL20), were quantified using quantitative real-time PCR (qRT-PCR). The expression levels were very similar between RNA-seq and qRT-PCR (Appendix A).

## 4. Discussion

Pathogenesis and immunity in host–pathogen interactions are complex phenomena that involve the host and pathogen in all stages of the disease process. Eimeria infection causes gastrointestinal diseases, resulting in serious economic damage in the poultry industry. Due to limitations in the use of anticoccidial agents because of issues such as anticoccidial drug resistance and drug residues in meat, it is time to find alternatives to address losses caused by coccidial infection. *E. acervulina* is a major *Eimeria* species in poultry farms that invades the duodenum. However, relatively little is known about host protective immunity compared to other *Eimeria* parasites. Thus, this study focused on transcriptomic analysis, along with the assessment of three parameters (intestinal lesion score, fecal oocyst production, and histopathological analyses), as indicators to measure the degree of the anticoccidial effect in berberine treatment of *E. acervulina*-infected chickens.

In vivo studies have demonstrated the positive efficacy of berberine in reducing fecal oocyst shedding in chickens infected with *E. tenella* [25,31]. Recently, berberine was shown to reduce oocyst production in chickens infected with five species of *Eimeria*, including *E. acervulina*, resulting in differences in sensitivity to fecal oocyst production. However, berberine was not found to have a direct effect on the sporulation of fecal oocysts [15]. After infecting broilers with *Clostridium perfringens* (2 × 10^8^ CFU) for 7 days, berberine administration in drinking water significantly reduced the intestinal lesion score and bacterial burden of *C. perfringens* on cecum microflora compared to the infected and untreated positive group. The intestinal lesion score of the berberine-treated group was similar to that of normal chickens [32]. Similarly, this study shows that fecal oocyst production and lesion scores were significantly reduced in chickens in the berberine-treated and *E. acervulina*-infected (EA+BBR) group compared to the untreated and infected (EA) group.

In the histopathological analysis in this study, *E. acervulina* infection significantly increased the crypt depth (CD) and decreased the villus height (VH). The EA+BBR group had a similar VH compared to the NC group. However, the CD of the BBR+EA group was similar to that of the EA group. These results indicate that berberine positively influences duodenum morphology by reducing intestinal lesion scores and fecal oocyst production and by increasing VH in chickens infected with *E. acervulina*. Interestingly, in normal healthy chickens fed a supplemented diet of 1 g berberine/kg for 12 or 21 days after hatching, berberine supplementation during 12 days significantly increased VH and increased the VH/CD ratio in the duodenum, but not in the jejunum. Meanwhile, berberine supplementation for 21 days did not increase VH in both tissues [33]. Berberine treatment significantly reduced fecal oocyst shedding and cecal lesion scores in chickens infected with 5 × 10^4^ sporulated *E. tenella* oocysts compared to untreated and infected chickens [26]. Broilers infected with *C. perfringens* and treated with berberine had significantly increased VH and VH/CD ratio in the duodenum, but not in both the jejunum and ileum, compared to the infected/untreated positive group or the normal healthy group used as a negative control. There were no significant differences in the lengths of CD in the three regions of the small intestine among the healthy, infected, uninfected, and treated groups [32]. Similarly to chickens, intraperitoneal administration of berberine to normal or radiation-injured mice significantly increased small intestinal length, as well as VH in all small intestinal segments, including the duodenum, jejunum, and ileum. For CD, no difference or a significant increase was observed depending on the small intestine sites [34]. On the other hand, CD was found to be either no different or significantly reduced in berberine-treated chickens compared to normal chickens, depending on the small intestine sites and berberine treatment period [33]. In addition, destruction of intestinal tissues in chickens infected with *Eimeria* caused growth retardation and impaired absorption of feed and minerals [35,36]. In general, it is known that the mineral absorption rate was higher in the duodenum than in the distal part of the intestines in broilers [36]. Berberine had a positive effect on the recovery of copper sulfate and zinc chloride imbalance caused by *E. tenella* infection [17]. Taken together, it is thought that berberine plays a positive role in increasing the VH of the duodenum in both the presence and absence of pathogenic infections that cause enteric diseases, and had effects on homeostasis and the absorption of feed and minerals.

The life cycle of *E. acervulina* in the host encompasses asexual and sexual multiplication, with three development stages: schizogony, gametogony, and sporogony [4]. Distinct mRNA expression profiles in *E. acervulina*-infected chickens were revealed according to development stage rather than strain variation [37], which induces complex host immune reactions. In our transcriptomic analysis using the duodenum from 2-week-old broilers infected with 1 × 10^5^ sporulated oocysts of *E. acervulina*, mucin2 and liver-enriched antimicrobial peptide 2 (LEAP20) mRNAs were decreased at 4 or 6 dpi. IFN-γ increased only at 6 dpi. Expression levels of TGF-β4, IL-2, IL-4, IL-6, and IL-18 mRNA were unchanged at 4 and 6 dpi (Appendix A). In support of these findings, IFN-γ and IL-8 mRNA were up-regulated in both lines at 4 days, but IL-4 mRNA was not increased in the duodenum in two lines (ROSS 308 and Hubbard JA 957) of 7-day-old broilers infected with 5 × 10^4^ sporulated oocysts of *E. acervulina* [38]. In 1-day-old broilers (ROSS 308) infected with 5 × 10^2^ and 5 × 10^4^ sporulated oocysts of *E. acervulina*, IL-8 mRNA was up-regulated in both inoculations, whereas IFN-γ, TGF-β2, TGF-β4, IL-2, IL-6, and IL-18 mRNA were not increased in the duodenum [39]. Three-week-old birds were orally challenged with 5 × 10^3^ sporulated oocysts of *E. acervulina* using two inbred chickens, SC and TK, with different susceptibilities to *E. acervulina* infection. IFN-γ mRNA expression in duodenal intraepithelial lymphocytes showed reduced levels in both strains. TGF-β4 mRNA levels were generally increased in duodenal lymphocytes in both strains [40]. In 3-week-old broilers (ROSS 708) infected with 1 × 10^5^ sporulated oocysts of *E. acervulina*, transcript levels of intestinal homeostasis-related genes (mucin2, avian beta-defensin 6 (AvBD6), LEAP2) were decreased in the duodenum at 5 and 7 dpi compared to uninfected chickens [41]. Kim et al. [42] used duodenal tissues from 3-week-old chickens infected with 1 × 10^4^ sporulated oocysts of *E. acervulina* and microarray chips and found that expression levels of IFN-γ increased only in mixed 3–4 dpi samples compared to mixed 1–2 dpi samples (days 1–2 vs. days 3–4). Taken together, immune responses during the innate phase of infection can be influenced by various factors, including age, breed, and dose of infection. Interestingly, transcriptomic analysis of duodenal tissues of 4-day-old Arbor Acres chickens infected with 5 × 10^5^ sporulated oocysts of precocious *E. acervulina* revealed that expression levels of IFN-γ, interferon regulatory factor 1, and IL-10 mRNA were significantly up-regulated in the very early timepoints (6, 18, 60, and 72 h) post-infection compared to uninfected controls [29].

In GO and KEGG pathway analyses comparing the DEGs of the EA group with those of the NC group, down-regulation of metabolism-related genes and up-regulation of cell integrity-related genes were generally observed at 4 dpi. At 6 dpi, an increase in genes related to the cytokine–cytokine receptor interaction term and cell homeostasis-related genes was generally observed. Twenty-five genes (Appendix A) associated with the cytokine–cytokine receptor interaction term after *E. acervulina* infection showed strong associations between IFN-γ and various chemokines in the PPI network at 6 dpi. These genes are involved in inflammatory pathways and immune response activation, which is very similar to the results of a previous transcriptome-based analysis of cecal tissues from chickens infected with three mixed *Eimeria* species (*E. acervulina*, *E. maxima*, and *E. tenella*) [43]. Additionally, transcripts of SLC7A5, IL1R2, GLDC, ITGB6, ADAMTS4, IL1RAP, TNFRSF11B, IMPG2, WNT9A, and FOXF1, which are associated with pivotal roles in the immune response, were identified in the jejunum of *E. maxima*-infected chickens [44].

One of the many functions of berberine is to protect against parasite infections, particularly its anticoccidial activity [16,25,26]. Our previous study also showed that berberine treatment reduced fecal oocyst production in chickens infected with five different *Eimeria* species, with different sensitivities depending on the species. However, berberine did not directly negatively affect oocyst sporulation [15]. In this study, berberine treatment not only had a positive effect on duodenal VH, but also reduced fecal oocyst production and intestinal lesion scores. Given that cellular immunity is an important factor in the defense against coccidiosis, it is likely that berberine treatment affected the host’s innate immune responses. In KEGG pathway analysis using DEGs at 4 dpi, the cytokine–cytokine receptor interaction term in the EA group was not included in the top 10 terms, although the term included eleven DEGs, including five up-regulated genes and six down-regulated genes. On the other hand, the cytokine-cytokine receptor interaction in the EA+BBR group was identified as the second term in the KEGG pathway analysis, and the term consisted of eleven DEGs, including ten up-regulated genes and one down-regulated gene. These results suggested a possibility that berberine treatment of infected chickens may have played a role in increasing protective immunity by affecting the order of gene expression toward genes related to the cytokine–cytokine receptor interaction term. Berberine-administered *C. elegans* mice showed a significant decrease in the bacterial burden of *Pseudomonas aeruginosa* compared to their untreated controls by enhancing innate immunity, primarily through the p38 MAPK pathway [45]. Berberine showed anti-inflammatory and immunomodulatory properties in ameliorating inflammatory bowel disease by targeting the Th1 and Th17 axis [46]. It is also worth noting that RNA-Seq analysis of two lines, resistant (C.B12) or susceptible (15I) to *E. maxima* infection, revealed differences in the expression kinetics of the cytokine genes between the two lines. Expression levels of IFN-γ and IL-10 transcripts were increased earlier in the jejunum in the resistant line compared to the susceptible line [47]. Taken together, it is thought that berberine treatment of chickens infected with *E. acervulina* may alleviate the pathogenesis of *Eimeria* infection by influencing immune responses similar to that of chickens resistant to *Eimeria* infection, but extensive follow-up studies are needed to further support this conclusion.

## 5. Conclusions

*E. acervulina* is one of the most prevalent strains in poultry farms, and is an important enteric disease that causes severe weight loss in broilers, resulting in serious economic losses for farms. Currently, the use of anticoccidial agents is restricted in farms. In order to control coccidiosis, it is important to understand the host’s protective immunity against *Eimeria* infection and to determine the host’s responses to natural products that can be used as alternatives to anticoccidial agents.

In this study, a berberine-based diet fed to chickens infected with *E. acervulina* resulted in a significant reduction in fecal oocyst production and intestinal lesion scores, an increase in duodenal villus height, and changes in intestinal immunity. Chickens are animals with physiological and structural differences from mammals, on which much research on berberine has been conducted. Thus, extensive follow-up studies are needed to understand the exact mechanism by which berberine is involved in intestinal immune activation. These findings suggest that berberine may have therapeutic and/or prophylactic potential in protecting hosts infected with *Eimeria*, an infectious and economically important disease.

## Figures and Tables

**Figure 1 biomolecules-15-00985-f001:**
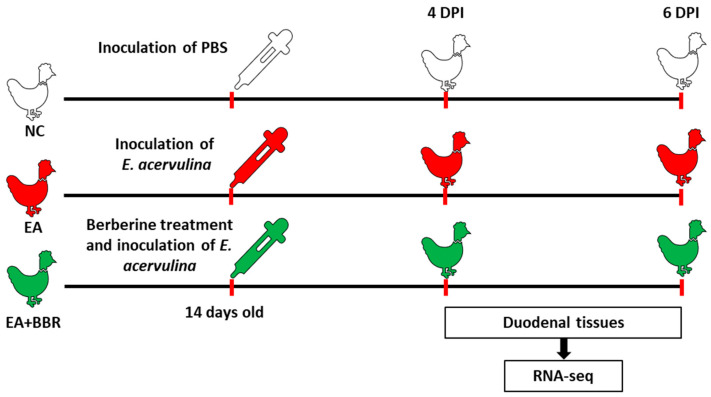
Schematic diagram of experiment design. Duodenal tissues were collected in berberine-treated and *E. acervulina* (EA)-infected broilers. Two-week-old ROSS 308 female chickens were orally infected with 1 × 10^5^ sporulated oocysts of EA. NC, uninfected control; EA, *E. acervulina* infection; EA+BBR, berberine treatment and inoculation of *E. acervulina*; DPI, days post-infection.

**Figure 2 biomolecules-15-00985-f002:**
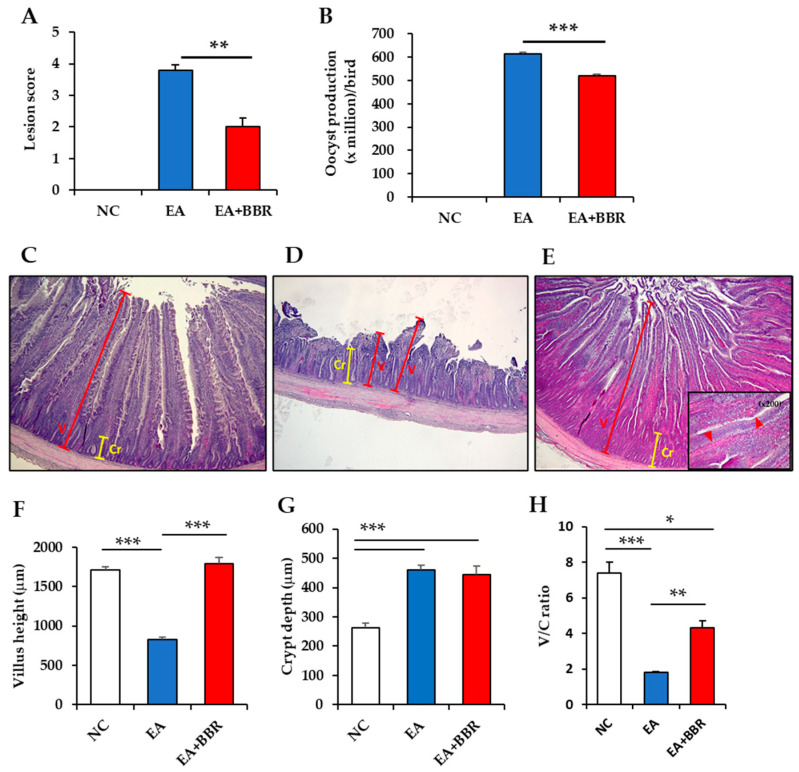
Comparison of clinical and histological symptoms in broilers after berberine treatment and *E. acervulina* (EA) infection. (**A**) Chickens (*n* = 7–9) were randomly selected for gut lesion scoring at 6 days post-infection. (**B**) Fecal oocyst shedding per bird (*n* = 30) was collected from days 5 to 9 post-infection. (**C**–**E**) Histopathologic lesions in the duodenum of uninfected control (**C**), EA-infected (**D**), and EA-infected chickens with berberine treatment (**E**). Five microscopic fields of each sample were randomly selected to measure villus height and crypt depth. Images are shown at 40× magnification and are representative of the five chickens in each group. Arrowheads in the inset (**E**) indicate oocysts of EA. Villus height (**F**), crypt depth (**G**), and villus vs. crypt ratio (*V*/*C*) (**H**) in each group. Student’s *t*-test was used for lesion score and oocyst production. A nonparametric statistical test using the Kruskal–Wallis test was used, and means were compared using Dunn’s multiple comparison test for (**F**–**H**). Data are expressed as the mean ± standard error values for one of two or three independent experiments that produce similar results. Differences were considered statistically significant at *p* < 0.05. NC, uninfected control; EA, *E. acervulina*; BBR, berberine; V, villi; Cr, crypt. * *p* < 0.05, ** *p* < 0.01 and *** *p* < 0.001.

**Figure 3 biomolecules-15-00985-f003:**
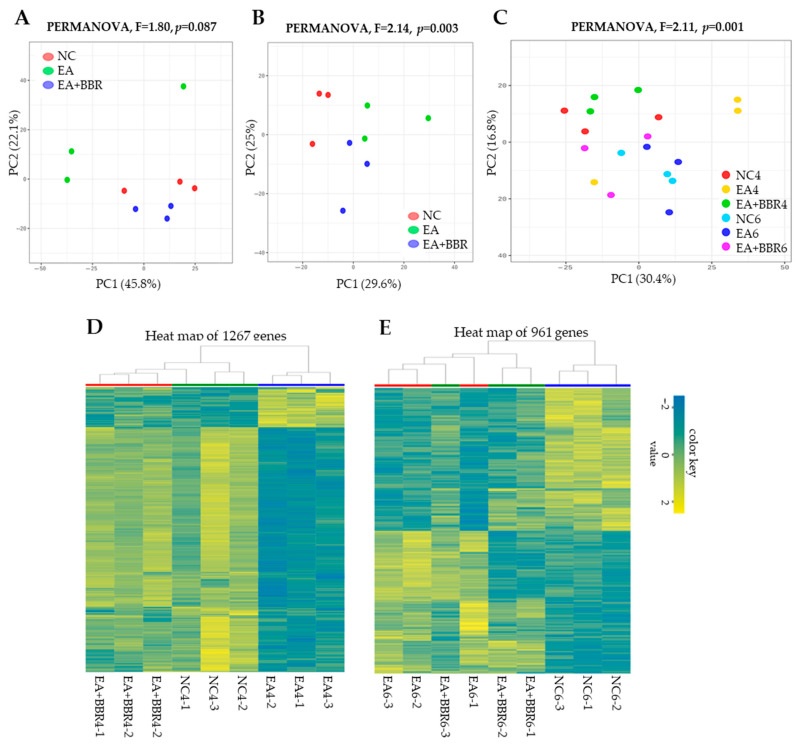
Transcriptomic data analysis of differentially expressed genes. (**A**–**C**) Plots of principal component analyses were based on expression level clustering of samples. *p*-value derived from the PERMANOVA test with 999 permutations. (**D**,**E**) Heatmap for hierarchical clustering between samples through clustering of the DEGs. Plots (**A**,**D**) are samples from day 4 post-infection. Plots (**B**,**E**) are samples from day 6 post-infection, and plot (**C**) includes both samples from days 4 and 6 post-infection. NC, uninfected control; EA, *E. acervulina*; BBR, berberine; PERMANOVA, permutational multivariate analysis of variance; DEGs, differentially expressed genes.

**Figure 4 biomolecules-15-00985-f004:**
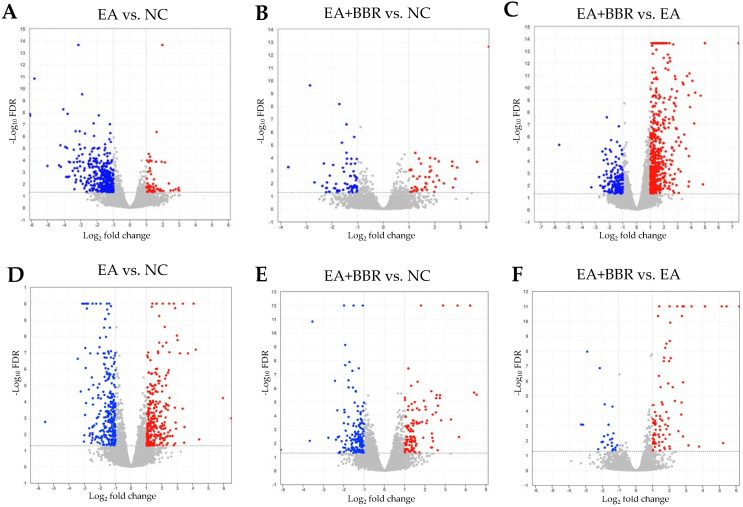
Volcano plots of differentially expressed genes. Plots (**A**–**C**) are samples from day 4 post-infection, and plots (**D**–**F**) are samples from day 6 post-infection. NC, uninfected control; EA, *E. acervulina*; BBR, berberine. Each dot represents a gene. Blue and red dots indicate down-regulated and up-regulated genes, respectively.

**Figure 5 biomolecules-15-00985-f005:**
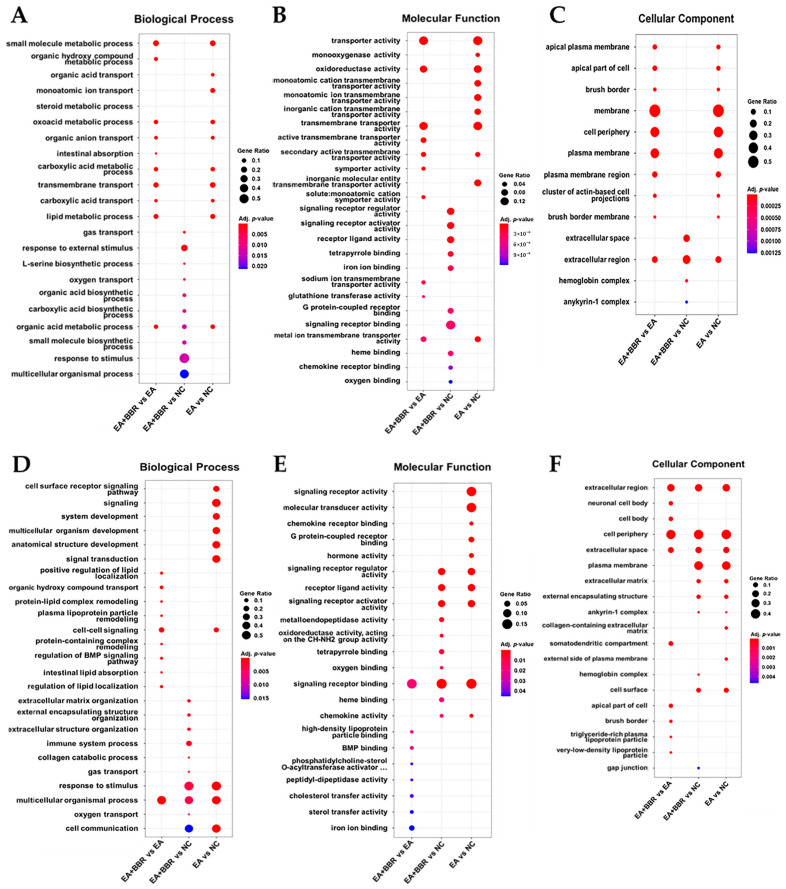
Top 10 ranked GO terms from GO enrichment analysis of DEGs. Plots (**A**–**C**) are samples from day 4 post-infection, and plots (**D**–**F**) are samples from day 6 post-infection. NC, uninfected control; EA, *E. acervulina*; BBR, berberine; GO, gene ontology; DEGs, differentially expressed genes.

**Figure 6 biomolecules-15-00985-f006:**
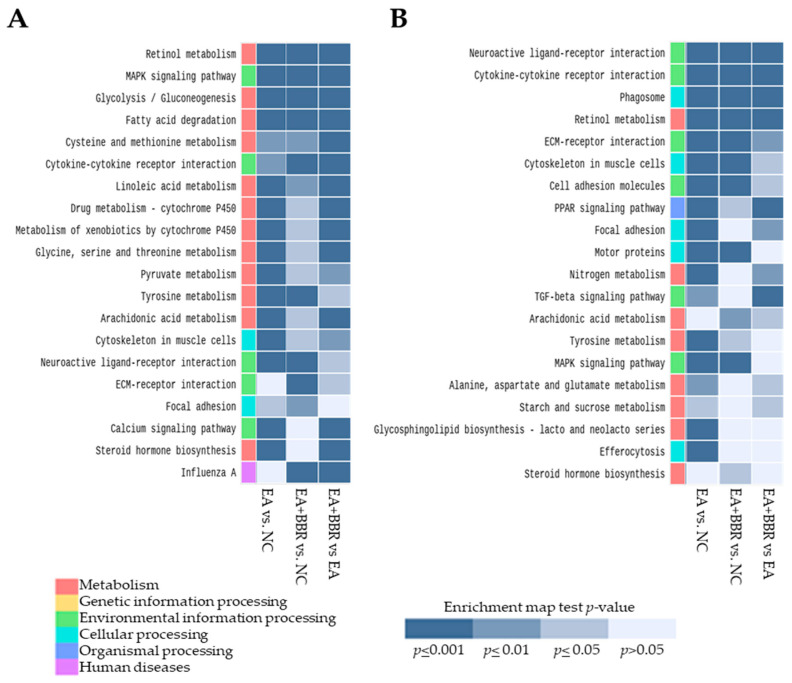
Top 20 ranked KEGG pathways from the KEGG enrichment analysis of DEGs. Plots (**A**) are samples from day 4 post-infection, and plots (**B**) are samples from day 6 post-infection. NC, uninfected control; EA, *E. acervulina*; BBR, berberine; KEGG, Kyoto Encyclopedia of Genes and Genomes; DEGs, differentially expressed genes.

**Figure 7 biomolecules-15-00985-f007:**
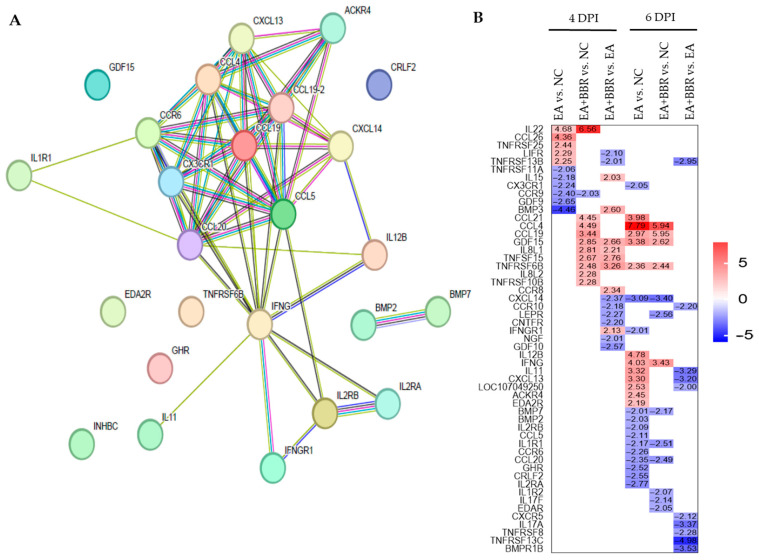
The protein–protein interaction (PPI) network and heatmap in cytokine–cytokine receptor interactions. (**A**) DEGs in the cytokine–cytokine receptor interaction term were obtained between groups (EA vs. NC) at 6 dpi. PPI analysis was performed using STRING. (**B**) Heatmap in cytokine–cytokine receptor interactions. The numbers in square boxes represent the expression levels of DEGs filtered by log2 fold change ≥2, *p*-adj false discovery rate <0.05. DPI, day post-infection; NC, uninfected control; EA, *E. acervulina*; BBR, berberine; DEGs, differentially expressed genes.

**Table 1 biomolecules-15-00985-t001:** Numbers of differentially expressed genes (DEGs) at 4 and 6 dpi in the duodenum of chickens infected with *E. acervulina*.

Days	Numbers of Genes Qualifying the Criteria	Total DEGs	Up-Regulated DEGs	Down-Regulated DEGs	Groups
4 DPI	15,408	441	62	379	EA vs. NC
		112	47	65	EA+BBR vs. NC
		714	534	180	EA+BBR vs. EA
6 DPI	15,486	572	286	286	EA vs. NC
		277	118	159	EA+BBR vs. NC
		112	87	25	EA+BBR vs. EA

Significant DEGs were filtered by log2 fold change ≥2, *p*-adj false discovery rate <0.05. DPI, day post-infection.

## Data Availability

The data presented in this study are available in the NCBI Sequence Read Archive, PRJNA1231613.

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
