# Peer review of "Berberine Reveals Anticoccidial Activity by Influencing Immune Responses in *Eimeria acervulina*-Infected Chickens"

_biomolecules, 2025, doi:10.3390/biom15070985_

Round 1

Reviewer 1 Report

Comments and Suggestions for Authors

The manuscript studied the effects of berberine on eimeria acervulina-infected intestinal tissue in chickens through transcriptome sequencing. The manuscript is interesting, but some major corcerns are listed. For example, growth performance such as body weight and feed intake, composition of basal diet were lacking. The transcriptome sequencing samples were less and not enough, only 3 samples in each treatment. Also, the differential genes involved in key immune-related pathways need to be verified and quantified by qPCR. Thus, I donot think the manuscript can be accepted in this version.

Author Response

We sincerely appreciate the editor and reviewers for their valuable time, and insightful recommendations for enhancing our manuscript. The authors have diligently considered the comments, and below are our point-for-point responses. We trust these will align with your expectations and contribute to the further improvement of our manuscript.

 Reviewer’s comments:

Comment: The manuscript studied the effects of berberine on Eimeria acervulina-infected intestinal tissue in chickens through transcriptome sequencing. The manuscript is interesting, but some major concerns are listed. For example, growth performance such as body weight and feed intake, composition of basal diet was lacking. The transcriptome sequencing samples were less and not enough, only 3 samples in each treatment. Also, the differential genes involved in key immune-related pathways need to be verified and quantified by qPCR. Thus, I do not think the manuscript can be accepted in this version.

Response: Thank you for your valuable comment. For body weight gain, the new sentences “Body weight gain was recorded at day 9 after post infection” and “Body weight gain was significantly decreased in EA+BBR group compared to NC and EA groups, but no difference was observed between NC and EA groups (Figure S1)” were added in lines 141 and 208, respectively. The authors did not have sufficient knowledge about feed intake. Therefore, no information on feed intake was available in this study. We agree that three samples per treatment is not sufficient. We thought that analyzing trends over multiple time points would be more helpful in understanding the effects of berberine treatment than analyzing a single specific time point. Therefore, samples were analyzed at two time points, 4 and 6 days after Eimeria infection and berberine treatment. We will do our best to follow your suggestions in further studies.  

About composition of basal diet, the new sentence “The composition of the broiler feed is provided (Table S1)” was added in line 123. Expression of genes involved in key immune-related pathways was quantified using qRT-PCR and compared with RNA-seq results. Thus, the sentence “The expression levels of three mRNAs, IL-22, IFN-γ and C-C motif chemokine ligand 20 (CCL20) were quantified using quantitative real-time PCR (qRT-PCR). The expression levels were very similar between RNA-seq and qRT-PCR (Table S7)” was added in line 364. The Supplementary Figure and Tables are added in the revised version as below.

Reviewer 2 Report

Comments and Suggestions for Authors

In this study, the authors evaluated berberine’s efficacy to the Eimeria acervuline infected chickens and attempted to elucidate its mechanism. The berberine treatment was successfully controlled Eimeria infection. However, the RNA-seq analysis revealed many changes on gene expression, and it seems the authors’ results are not conclusive. The authors could strengthen their manuscript by addressing the following concerns.

Major comments

  1. The interpretation of the KEGG pathway analysis appears subjective. The authors observed that many pathways changed after infection and berberine administration, but they concluded that changes in innate immunity contributed to the control of infection. While RNA-seq analysis indicated changes in immune-related genes, there is no direct evidence that these gene expression changes contributed to disease control. Do the authors agree or disagree that transporter activity or other pathways have influenced the disease outcome? Additionally, while I agree that the immune response is complex, the conclusion of "activation of innate immunity" seems to contradict the berberine's general notion of "anti-inflammation." Although it may be outside the main focus, what about the potential contributions of microbiome changes to the disease?

  1. The control group for berberine treatment without parasite infection is missing. If the authors had included a group receiving berberine administration only, they could compare the results more clearly. Regarding the comparison between EA+BRR vs NC, there are two variable factors, making it difficult to interpret the data.

Other comments

  1. As I pointed out in major comment #1, the causal relationship between the disease control and immune stimulation has not been established. The authors should consider revising the title.

  1. In the figure legends, the term “normal control” suggests the existence of an abnormal control. Uninfected control (Line 207) seems more appropriate.

Author Response

We sincerely appreciate the editor and reviewers for their valuable time, and insightful recommendations for enhancing our manuscript. The authors have diligently considered the comments, and below are our point-for-point responses. We trust these will align with your expectations and contribute to the further improvement of our manuscript.

Reviewer’s comments:

In this study, the authors evaluated berberine’s efficacy to the Eimeria acervulina infected chickens and attempted to elucidate its mechanism. The berberine treatment was successfully controlled Eimeria infection. However, the RNA-seq analysis revealed many changes on gene expression, and it seems the authors’ results are not conclusive. The authors could strengthen their manuscript by addressing the following concerns.

Major comments

Comment 1: The interpretation of the KEGG pathway analysis appears subjective. The authors observed that many pathways changed after infection and berberine administration, but they concluded that changes in innate immunity contributed to the control of infection. While RNA-seq analysis indicated changes in immune-related genes, there is no direct evidence that these gene expression changes contributed to disease control. Do the authors agree or disagree that transporter activity or other pathways have influenced the disease outcome? Additionally, while I agree that the immune response is complex, the conclusion of "activation of innate immunity" seems to contradict the berberine's general notion of "anti-inflammation." Although it may be outside the main focus, what about the potential contributions of microbiome changes to the disease?

Response: Thank you for your valuable comment. The pathogenesis of E. acervulina infection was alleviated by berberine treatment, which is thought to be the result of many factors including transporter activity, immune regulation, and other pathways. Since berberine also has various biological activities, including antiarrhythmic, antiplatelet, antidiabetic, antiviral, antibacterial, and anti-inflammatory pharmacological effects, it is not easy to directly prove whether there is a correlation between the RNA-seq data obtained from Eimera-infected hosts and the role of immune-related genes in the alleviation of pathogenesis, as the reviewer commented. RNA-seq analysis results showed that the expression levels of immune-related genes in the berberine-treated group were detected earlier than in the berberine-untreated group. Although genes in other pathways may also be affected, the authors focused on immune-related genes. The authors fully agreed that transporter activity and/or other pathways have influenced the disease outcome.

In the section of conclusion and other sections, “activating innate immunity” was changed to “by influencing immune responses” in lines 505 and 516. The duodenal region has a less active microbiota compared to the ileum and cecum. In addition. berberine treatment significantly reduced oocyst production in chickens infected with E. tenella that infected the cecum (ref. 15). Therefore, the authors thought that the change in gut microbiota by berberine treatment had little effect on disease prognosis.

 Comment 2: The control group for berberine treatment without parasite infection is missing. If the authors had included a group receiving berberine administration only, they could compare the results more clearly. Regarding the comparison between EA+BRR vs NC, there are two variable factors, making it difficult to interpret the data.

Response: Compared to the well-studied and well-informed E. tenella studies on host response changes, information on E. acervulina infections has been very limited until recently.

Although this study has limitations, such as omitting the berberine-only group and the lack of direct evidence between berberine administration and symptom relief in Eimeria-infected chickens through RNA-seq analysis, authors believe that the information obtained from this work may be useful for further studies. In future studies, authors will fully consider and include the suggested points.

Other comments

Comment 3: As I pointed out in major comment #1, the causal relationship between the disease control and immune stimulation has not been established. The authors should consider revising the title.

Response: We thank the reviewer’s suggestion. The tile “Berberine reveals anticoccidial activity by activating immune responses in Eimeria acervulina-infected chickens” was changed to “Berberine reveals anticoccidial activity by influencing immune responses in Eimeria acervulina-infected chickens”

Comment 4: In the figure legends, the term “normal control” suggests the existence of an abnormal control. Uninfected control (Line 207) seems more appropriate.

Response: According to reviewer’s suggestion, “normal” control was changed to “uninfected” control throughout the manuscript, excepting the annotated “normal” from the published papers.

Round 2

Reviewer 2 Report

Comments and Suggestions for Authors

The authors have responded to the reviewer's concerns.